# Frequency of Convenience Cooking Product Use Is Associated with Cooking Confidence, Creativity, and Markers of Vegetable Intake

**DOI:** 10.3390/nu15040966

**Published:** 2023-02-15

**Authors:** Natasha Brasington, Tamara Bucher, Emma L. Beckett

**Affiliations:** 1School of Environmental and Life Sciences, The University of Newcastle, Ourimbah 2258, Australia; 2Hunter Medical Research Institute, New Lambton Heights 2305, Australia

**Keywords:** meal bases, recipe bases, convenience cooking products, vegetables, cooking sauces, cooking

## Abstract

Low levels of cooking skills, confidence and home cooking are related to poorer dietary outcomes and are a common barrier to adequate vegetable consumption. Convenience cooking products may play a role in lowering the levels of confidence and creativity required to prepare home-cooked meals. It has previously been reported that those who use convenience cooking products have lower levels of cooking confidence and creativity and lower vegetable intakes compared to those who do not use these products. However, the relationship between these outcomes and the frequency of use of convenience cooking products has not been assessed. Therefore, a balanced demographic panel of Australian adults (*n* = 1034) was surveyed on the frequency of convenience cooking product use, vegetable intake and variety, and opinions and habits regarding vegetable intake. Those who used the products more regularly had higher cooking confidence and creativity, and higher vegetable variety scores, compared to less regular users (*p* < 0.05). However, the frequency of use of convenience cooking products was not associated with higher vegetable intake and did not influence views around the ease of eating vegetables. Therefore, these products may be a tool for assisting those with lower levels of cooking skills in accessing a higher variety of vegetables, but vegetable quantity in these products may need to be revised to assist consumers in better meeting intake recommendations.

## 1. Introduction

Cooking skills and the facilitation of home cooking are important contributors to balanced diets [1,2]. However, there are numerous barriers to home cooking from basic or core food ingredients, including cooking skills, confidence and knowledge, time, costs, convenience, attitudes, mood and health [3]. Pressures such as both parties in households tending to work full-time hours [4], the ready availability and marketing of foods cooked out of the home [5] and a reduced emphasis on home economics training in education curricula are contributing to low levels of cooking skills and home cooking in many populations [6,7,8,9,10]. Low levels of cooking skills and home cooking are related to poorer dietary outcomes, with those who cook more often at home being more likely to consume healthy foods more often [11,12,13], and those who tend to have a high level of nutrition knowledge and cooking skills are more likely to achieve the recommended consumption amounts of fruit and vegetables daily [14].

Vegetable consumption is a crucial component, and a marker of, a healthy balanced diet. Vegetables typically require cooking skills to facilitate consumption and enhance palatability. When cooking skills and confidence are low, individuals are more susceptible to using and consuming convenience and commercial foods (takeaway/takeout, fast foods and ready-to-heat meals). Consumer behaviour models also show other attitudinal drivers of convenience food consumption such as guilt [15], moral motivations [16], “convenience orientation” towards time and energy savings [17,18] and time and budget perceptions. Conversely, health consciousness and enjoyment reduce engagement with convenience foods [18]. Recent trends demonstrate the normalisation of convenience foods and thus their importance in dietary patterns [15,19]. While convenience and commercial food products require little or no preparation, reducing the time needed to prepare meals [6], they are typically lower in nutritional quality and are more energy-dense [2,20].

However, some convenience food products are used to reduce the burden of barriers such time, skills and costs, while encouraging and facilitating cooking. These “convenience cooking products” are distinct from other convenience food products in that they are used as ingredients in cooking and can facilitate cooking, as they include back-of-pack recipes with suggested ingredients. This lowers the skill threshold, as well as reducing meal preparation and cooking times. Common convenience cooking products include meal and recipe bases (in liquid or powder form, commonly sold in retort pouches and typically including complete meal recipes), simmer sauces (sold in retort pouches or jars, typically including recipes for complete meals or meal components and serving suggestions) and pasta sauces (sold in jars with recipes or preparation instructions specifically to serve with pasta). Due to the natural comparison to the healthfulness of meals prepared from basic, core and whole ingredients (i.e., cooking from “scratch”), and their often concentrated form, they may be regarded as processed foods and therefore perceived as non-healthful [21,22,23]. However, it is possible that through the inclusion of healthful back-of-pack recipes (e.g., proposing the addition of vegetables), these products may have a positive impact on nutritional intakes and cooking skills.

We have previously reported, using a convenience sample of 842 Australians, that those who reported using convenience cooking products had lower scores for cooking confidence and cooking creativity than those who do not use them [24], suggesting that these products might be a tool to facilitate home cooking in those with low levels of confidence and skills, as users reported typically following the back-of-pack recipes. However, users also typically had lower vegetable intakes [25], and an audit of meal and recipe base products available in Australian supermarkets revealed low vegetable variety in the back-of-pack recipes [26], potentially suggesting that improvements in product design are needed to allow these products to be regarded as tools for facilitating healthy eating.

However, the sample previously studied was obtained via a snowball recruitment technique and, as such, was limited in size and scope, with a skewed demographic profile, without clear findings on frequency of use. Although users of these products may have lower cooking confidence and creativity than those who do not use them, these attributes may vary within users by frequency of use. We hypothesised that a more regular user would have higher confidence and creativity than those who used the same products but less frequently. Understanding these relationships will help better design recommendations and tools for people who opt for convenience cooking over more involved home cooking methods. Therefore, we surveyed a larger, representative (based on Australian national census data [27]) and balanced sample of Australians who were self-reported consumers of convenience cooking products on their usage habits, cooking confidence and creativity and markers of their vegetable intake to assess the relationships in users of key convenience cooking products (meal/recipe bases, simmer sauces and pasta sauces).

## 2. Materials and Methods

### 2.1. Study Design and Recruitment

This study was granted ethical approval (Human Research Ethics Committee, the University of Newcastle, Reference H-2020-0119). A cross-sectional survey was conducted during November 2020 utilising an online market research panel recruited via Qualtrics, (SAP, Provo, UT, USA) using probability sampling. during November 2020. Questions were organised thematically into blocks including questions on convenience cooking product usage habits, cooking skills (confidence and creativity), vegetable intake (serves per day and variety scores) and demographics. For inclusion, participants were required to be living in Australia, over 18 years of age and self-reported users of convenience cooking products (used any meal/recipe base, or commercially available cooking or simmer sauce, at least once a month). As the survey was in English, English comprehension skills were a practical inclusion requirement.

### 2.2. Frequency of Use

Current frequency of use of the convenience cooking products in the major categories of meal and recipe bases (or concentrates), simmer sauces and pasta sauces was recorded. Options for selection were “multiple times a week”, “once a week”, “once a month” and “less than once a month”. However, these responses were subsequently collapsed into 3 categories (>weekly, weekly and <weekly) for analysis.

### 2.3. Cooking Confidence Scores and Cooking Creativity Scores

Scores for cooking confidence and creativity were calculated using questions based on cooking identity and food creativity scales previously published [24]. The presented statements were rated on 5-point Likert scales ranging from strongly disagree (1) to strongly agree (5). Ratings were summed to calculate scores for cooking confidence (out of 35) and cooking creativity (out of 30). Reversed phrasing questions in these scales were used as attention checks, and those who failed were excluded from the analysis. Internal validity was assessed using Cronbach’s Alpha calculations.

### 2.4. Vegetable Intake and Variety Scores

Typical daily vegetable consumption (serves per day with a guide provided to assist with estimation) was self-reported. These data were then categorised into meeting or not meeting the recommendations for daily consumption of vegetables based on the Australian Guide to Healthy Eating. Vegetable variety scores were calculated based on the frequency of consumption of vegetables commonly consumed by the Australian population [24], with 1 point scored for each listed vegetable eaten at a frequency of one serve per week or more. Fourteen vegetables were listed, resulting in a maximum score of 14.

### 2.5. Vegetable Opinions, Behaviour and Knowledge

Participants were asked to rate how much they agree (5-point Likert scale from strongly disagree to strongly agree) with 3 statements regarding perceptions of vegetable eating (1. Eating 5 serves of vegetables each day is easy; 2. Eating a variety of vegetables each day is easy; 3. I like vegetables.). Participants were also asked to select how many serves or vegetables per day were recommended for good health (numeric scale), and they were asked how likely they were to follow recommendations such as serving with a side salad/vegetables or adding optional vegetable ingredients if recommended on the pack (5-point Likert scale from definitely not to definitely [18]).

### 2.6. Statistical Analysis

Data were analysed using JMP (Pro 14; SAS Institute Inc., Cary, NC, USA). Both categorical and continuous data were used. A *p*-value threshold of <0.05 was utilised for assessing statistical significance. Contingency tables (Pearson χ^2^) were used to investigate the relationships between categorical variables. Standard least squares regression was used to compare adjusted means by category with Tukey’s HSD post hoc tests. Analyses were also adjusted for age, sex, income, education, working hours and nights per week cooking at home.

## 3. Results

### 3.1. Participants and Demographics

After exclusions for incomplete responses, failure of attention checks and removal of those who completed the survey in less than half the median completion time, 1034 complete and valid responses were included in the present analyses. Participant age ranged from 18 to 88 years, with a median age of 43 years, and participants were relatively evenly distributed across age groups, education levels and income levels (Table 1). Participants were 52.8% females. The majority of participants were working 30 h or less per week (noting that this survey was conducted while Australia was impacted by the COVID-19 pandemic; Table 1). The majority of participants reported cooking dinner at home 5 days a week or more (Table 1).

### 3.2. Frequency of Use

The most common frequency of use was weekly for each convenience cooking product class assessed (meal and recipe bases, simmer sauce and pasta sauces; Table 2). Income, working hours, education level, frequency of cooking dinner at home, and age distributions varied by the frequency of use for each convenience cooking product class (Appendix A).

### 3.3. Cooking Confidence and Creativity

High internal reliability of the scales used for cooking confidence and creativity was found, with calculations returning Cronbach’s Alpha scores of 0.90 and 0.88, respectively. Scores for cooking confidence ranged from 7 (the minimum possible score) to 35 (the maximum possible score) with a mean of 25.1 (standard deviation 5.6). Scores for cooking creativity ranged from 6 (the minimum possible score) to 30 (the maximum possible score), with a mean of 18.5 (standard deviation 4.9).

Mean cooking confidence scores were higher in those who used each of the convenience cooking products most regularly (Figure 1A–C) compared to those who used them less frequently. These results remained significant when analyses were conducted with adjustments for age, sex, income, education, work hours and cooking frequency applied (Appendix A). Mean cooking creativity scores were higher in those who used each of the convenience cooking products most regularly compared to those who used them less frequently (Figure 1D–F). These results remained significant when adjustments for age, sex, income, education, work hours and cooking frequency were applied to analyses (Appendix A).

When analyses were stratified by sex, confidence scores by frequency of meal and recipe base use differed only in males, with the exception of the use of meal and recipe bases, which showed similar patterns in males and females (Figure 2). Interaction terms for sex and frequency of use were significant for meal and recipe bases (*p* = 0.02), simmer sauces (*p* = 0.008) and pasta sauces (*p* = 0.01) in predicting confidence scores. Mean confidence scores did not differ (*p* = 0.6) between males (mean 25.3, standard deviation 5.7) and females (mean 25.1, standard deviation 5.5). The results remained similar when adjustments for age, sex, income, education, work hours and cooking frequency were applied (Appendix A).

Creativity scores were higher in males (mean 19.1 standard deviation 4.8) than in females (mean 18.1 standard deviation, 4.9, *p* = 0.002). When analyses were stratified by sex, creativity scores by frequency of meal and recipe base use differed only in males, with the exception of the use of simmer sauces, which showed similar patterns in males and females (Figure 2). Interaction terms for sex and frequency of use were significant for meal and recipe bases (*p* = 0.04) and pasta sauces (*p* = 0.04) in predicting confidence scores, but not for simmer sauces (*p* = 0.09). These results remained similar when adjustments were applied for age, sex, income, education, work hours and cooking frequency (Appendix A).

### 3.4. Vegetable Intake

The consumption of five or more serves of vegetables per day was reported by 5.4% of the sample. Frequencies of use of meal and recipe bases (χ^2^ = 0.7, *p* = 0.7), simmer sauces (χ^2^ = 0.7, *p* = 0.7) and pasta sauces (χ^2^ = 0.4, *p* = 0.8) were not associated with differences in the proportion of participants eating at least five serves of vegetables per day. However, vegetable variety scores were associated with frequency of use, with vegetable variety score increasing with frequency of use in all product categories (Figure 3). There was no interaction between sex and frequency of use in predicting vegetable variety score for meal and recipe bases (*p* = 0.8), simmer sauces (*p* = 0.7) or pasta sauces (*p* = 0.6). These results remained similar when analyses were stratified by sex and were adjusted for age, sex, income, education, work hours and cooking frequency (Appendix A). In a multifactorial model, frequencies of use of each of the convenience cooking products (meal/recipe bases, simmer sauces and pasta sauces) were significant predictors of vegetable variety scores (Standardised Beta = 0.1, 0.04, 0.06, respectively, *p* < 0.05), along with age (0.1), income (0.1) and frequency of cooking (−0.08).

The largest proportion (33.0% and 31.5%) of respondents gave a neutral rating to the statements “Eating 5 serves of vegetables each day is easy” and “Eating a variety of vegetables each day is easy”. Strongly agree (40.0%) was the most common response to “I like vegetables”. Convenience cooking product frequency of use was not related to any of these self-reported opinions on vegetable intake (Table 3, all *p* > 0.05).

When asked how many serves of vegetables per day were recommended for good health, 33.1% responded correctly (approximately five serves), 22.1% overestimated requirements and 44.8% underestimated requirements. Those who use meal and recipe bases more than weekly were more likely (χ^2^ = 14.2, *p* = 0.0007) to overestimate needs (31.1%) compared to those who used them weekly (16.9%) or less than weekly (21.2%). Distributions did not vary by frequency of use of simmer sauces or pasta sauces (*p* > 0.05).

When asked if participants follow serving suggestions such as “serve with salad” or “serve with vegetables” when using convenience cooking products “might or might not” was the most common response at 46%. Thirty-eight percent were “likely” to follow serving suggestions (28% probably, 10% definitely), and only 17% were “not likely” to (3% definitely not, 14% probably not). When asked if they would add optional extra ingredients if suggested on the pack, “might or might not” was the most common response at 50%. Thirty-eight percent were “likely” to include optional ingredients (31% probably, 7% definitely), and only 11% were “not likely” to (2% definitely not, 9% probably not). These distributions did not vary by frequency of use of any convenience cooking product category.

## 4. Discussion

This study is the first to investigate the relationship between the frequency of convenience cooking product use and outcomes related to cooking confidence, cooking creativity and vegetable intakes. These findings extend upon previous research studying users of these products, compared to non-users, without consideration of the frequency of use [25]. The findings here regarding cooking creativity and confidence complement the earlier findings that found use was linked to higher cooking confidence and creativity, with more regular use of convenience cooking products (meal/recipe bases, simmer sauces and pasta sauces) being associated with higher scores. Here, the frequency of use correlated with vegetable variety scores, with more frequent use linked to higher variety, which is interesting given there was no difference in variety scores between users and non-users previously [25]. However, this did not result in a higher overall vegetable consumption level.

This suggests that these convenience cooking products may be vehicles for enhancing cooking skills and, either directly or indirectly, introducing vegetable variety into the diet through encouraging a wider variety of recipe use and meal creation or through increasing the confidence and creativity required to incorporate vegetables. Research suggests that consuming a wide variety of plant-based foods daily will provide health benefits [28]. Different sources of vegetables provide the body with a variety of sources of micronutrients and phytochemicals for health [29]. However, the low overall vegetable intake across all frequency-of-use groups may suggest that these products do not contain sufficient serves of vegetables in their back-of-pack recipes [26]. Only 5.4% of the total cohort reported consuming five or more serves of vegetables per day. This reflects the findings of the 2018 Australian National Health Survey which found that 95% of Australians do not consume the recommended five or more serves of vegetables daily [30]. This suggests that the following study is likely to be a good sample of the Australian adult population.

The findings regarding opinions on vegetable intake suggest that convenience cooking products may be influencing vegetable consumption and confidence, directly leading people to consider the challenges associated with vegetable consumption, as opinions on the ease of eating enough and a variety of vegetables and whether participants reported liking vegetables did not vary by frequency of convenience cooking product use. Regular meal/recipe base users were more likely to overestimate how many vegetables are recommended for health; this may be due to higher exposure to back-of-pack recipes or may be a variable that encourages higher use of these products.

The relationships for cooking confidence and creativity and frequency of convenience cooking product use were due to significant relationships in males, with limited relationships found in females. While women are more likely to take responsibility for home cooking [31], men’s motivation for cooking more highly correlates with cooking enjoyment [31], with men likely to acquire cooking skills when cooking is considered to be an enjoyable activity, in comparison to females whose familiar roles are mostly characterised by their task as the main food provider and for whom cooking tends to be an obligation. Convenience cooking products may assist with skill development by lowering the threshold of effort required, which may make cooking a more enjoyable activity. Older, single men have previously been found to consume fewer fruits and vegetables than married men, and men were also less likely to cook a range of meals and more likely to choose foods that were easy to prepare than women of the same age group [32]. Therefore, these products may have specific appeal to males who have lower cooking skills and creativity.

Importantly patterns were generally the same across each subcategory of convenience cooking products. This may indicate high concurrent use or may indicate that these product categories fulfil similar roles for users. While there are likely variations in the vegetable contents of these products and their back-of-pack recipes, empirical data on these differences are needed to further understand these relationships.

Limitations of this study and other directly related studies are that data have been collected by self-reported survey means, and only in a cross-sectional manner. Therefore, no causal insights can be gained into relationships with health status or into how confidence, creativity or vegetable variety would vary when these products are introduced or removed from the diet. It is possible that convenience cooking products reduce the need for pre-prepared convenience foods and take-away products. However, it is also possible that these products displace cooking from core ingredients. Strengths of this study include the utilisation of a market research panel to obtain a large sample with key demographics representative of the broader Australian population as per Australian Census data. This is the first study regarding convenience cooking product use in a representative sample, and these data are important for addressing the information gap on these frequently consumed products.

## 5. Conclusions

The objective of this study was to assess the relationships between convenience cooking product usage habits, cooking confidence and creativity, vegetable intake and attitudes. The findings presented here describe a nascent research field in convenience cooking products and suggest that rather than convenience cooking products being unhealthy processed foods, they may have a role as a public health tool. Furthermore, this research has a practical utility in that these products have the potential to be recommended by dietitians and other health professionals for consumers to utilise their current cooking skills and confidence when cooking in the kitchen. The knowledge generated here may also inform the future design of these products to appeal to both consumer needs and nutritional needs. More frequent use is associated with higher confidence, creativity and vegetable variety, so despite being potentially viewed as processed and not as good as cooking from core food ingredients, convenience cooking products might be beneficial tools for those with low skills and other barriers such as time and costs. More research is needed into the role convenience cooking products do and may be able to play in encouraging higher vegetable intakes for health, including longitudinal research and intervention trials to demonstrate causation.

## Figures and Tables

**Figure 1 nutrients-15-00966-f001:**
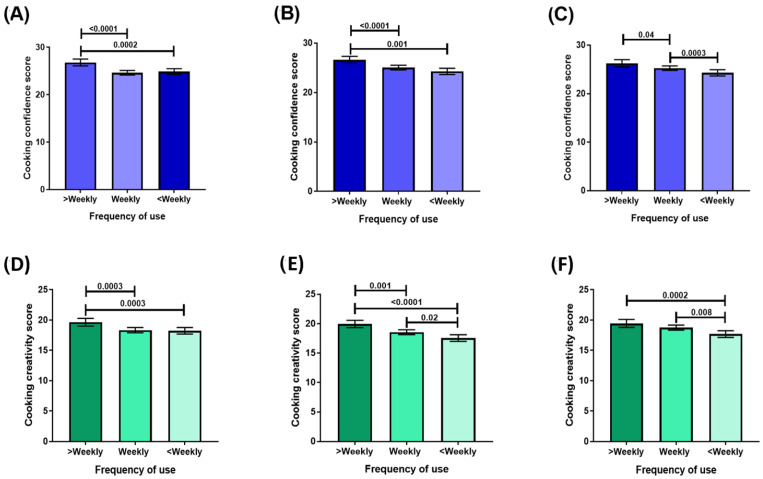
Cooking confidence scores by frequency of use of convenience cooking products: (**A**) meal and recipe bases; (**B**) simmer sauces; (**C**) pasta sauces. Cooking creativity scores by frequency of convenience cooking product use: (**D**) meal and recipe bases; (**E**) simmer sauces; (**F**) pasta sauces.

**Figure 2 nutrients-15-00966-f002:**
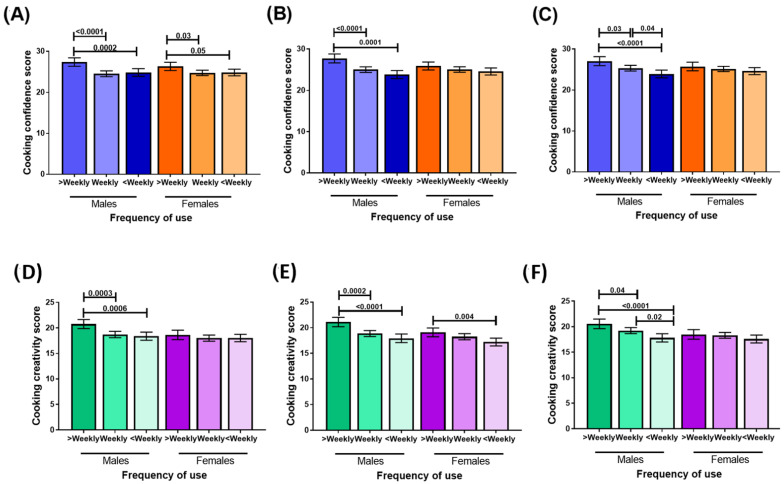
Cooking confidence scores by frequency of use of convenience cooking products, stratified by sex: (**A**) meal and recipe bases; (**B**) simmer sauces; (**C**) pasta sauces. Cooking creativity scores by frequency of convenience cooking product use, stratified by sex: (**D**) meal and recipe bases; (**E**) simmer sauces; (**F**) pasta sauces.

**Figure 3 nutrients-15-00966-f003:**
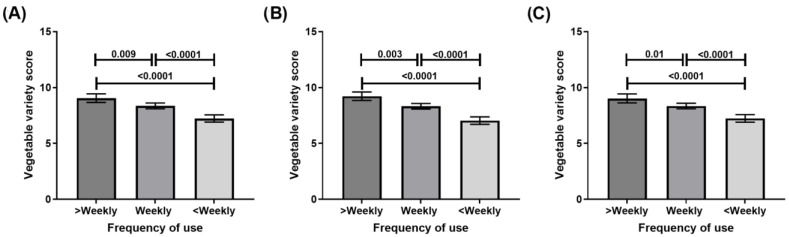
Vegetable variety scores by frequency of convenience cooking product use: (**A**) meal and recipe bases; (**B**) simmer sauces; (**C**) pasta sauces.

**Table 1 nutrients-15-00966-t001:** Demographics characteristics of sample.

Variable	*n* (%)
Sex	
Male	480 (46.4)
Female	546 (52.8)
Others	8 (0.1)
Income	
AUD < 35,000	188 (18.1)
AUD 35,000–AUD 49,999	154 (14.9)
AUD 50,000–AUD 74,999	188 (18.1)
AUD 75,000–AUD 149,999	150 (14.5)
AUD 150,000–AUD 199,999	218 (21.1)
AUD > 200,000	55 (5.3)
Declined to answer	81 (7.8)
Hours worked/week	
<15	464 (44.9)
15–30	195 (18.9)
30–50	351 (33.9)
50+	24 (23.2)
Education	
<Year 12 ^#^	134 (13.0)
Year 12 ^#^	207 (20.0)
Technical diploma	300 (29.0)
Bachelor’s degree	255 (24.7)
Postgrad degree	127 (12.3)
Declined	11 (1.1)
Nights/week cooking at home	
>7	384 (37.1)
5–6	313 (30.2)
3–4	247 (23.9)
1–2	85 (8)
<1	5 (0.4)
Ages	
18–24	163 (15.8)
25–34	190 (18.4)
35–44	173 (16.8)
45–54	160 (15.5)
55–65	153 (14.8)
65+	192 (18.6)

^#^ final year of secondary education in Australia.

**Table 2 nutrients-15-00966-t002:** Reported usage frequency of selected convenience cooking products.

	Meal and Recipe Bases		Simmer Sauces		Pasta Sauces	
	*n*	%	*n*	%	*n*	%
Multiple times a week	227	21.3	229	22.1	207	20.0
Weekly	494	47.9	519	50.2	533	51.5
Monthly or less	313	30.3	286	27.7	294	28.4

**Table 3 nutrients-15-00966-t003:** Correlation between vegetable eating opinions and convenience cooking product frequency of use.

	Frequency of Use (χ^2^(*p*))		
Statement	Meal/Recipe Bases	Simmer Sauces	Pasta Sauces
Eating 5 serves of vegetables each day is easy	0.86 (0.4)	6.8 (0.6)	13.3 (0.1)
I like vegetables	14.6 (0.07)	13.1 (0.1)	6.2 (0.6)
Eating a variety of vegetables each day is easy	6.4 (0.6)	11.2 (0.2)	7.4 (0.5)

## Data Availability

Data are available upon request (pending ethics approvals).

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
