# Peer review of "Frequency of Convenience Cooking Product Use Is Associated with Cooking Confidence, Creativity, and Markers of Vegetable Intake"

_nutrients, 2023, doi:10.3390/nu15040966_

Round 1
Reviewer 1 Report
The study examines a current issue, a relevant consumer trend.
The chosen topic offers the possibility of a number of literature connections, in this respect I suggest extending the range of the literature covered.
Firstly, a more nuanced presentation of convenience consumption in line with consumer behaviour models.
In addition, I propose a sophisticated presentation of the vegan lifestyle in relation to the trend of health consciousness, with a focus on the lifestyle-values-food consumption behaviour nexus.
As for the methodology, I propose to extend it by presenting the methodological context of the research aim, hypotheses, results, the research model and the conceptualization process.
Regarding the measurement methods, I propose to explain the questionnaire questions in relation to the literature conclusions and the measurement techniques used (Likert scale), explaining the reason for the choice of the unpaired scale.
I propose to extend the conclusions of the research with a summary of the research objectives and hypotheses and a comprehensive description of the practical utility of the research.
Author Response
As for the methodology, I propose to extend it by presenting the methodological context of the research aim, hypotheses, results, the research model and the conceptualization process.
Regarding the measurement methods, I propose to explain the questionnaire questions in relation to the literature conclusions and the measurement techniques used (Likert scale), explaining the reason for the choice of the unpaired scale.
- The Likert Scales for cooking skills and confidence were adopted as per the scales as referenced. The Likert scales used for vegetable opinions are standard scales used for agreement.
I propose to extend the conclusions of the research with a summary of the research objectives and hypotheses and a comprehensive description of the practical utility of the research.
- We have extended to conclusion to now read “The objective of this study was to assess the relationships between convenience cooking product usage habits, cooking confidence and creativity, vegetable intake and attitudes. The findings presented here are describe a nascent research field in convenience cooking products, and suggest that rather than convenience cooking products being unhealthy processed foods, they may have a role as a public health tool. Furthermore, this research has a practical utility in that these products have the potential to be recommended by dietitians and other health professionals for consumers to utilise their current cooking skills and confidences when cooking in the kitchen. The knowledge generated here may also inform the future design of these products to appeal to both consumer needs and nutritional needs. More frequent use is associated with higher confidence, creativity, and vegetable variety so despite being potentially viewed as processed and not as good as cooking from core food ingredients, convenience cooking products might be beneficial tools for those with low skills and other barriers like time and costs. More research is needed into the role convenience cooking products do and may be able to play in encouraging higher vegetable intakes for health, including longitudinal research and intervention trials to demonstrate causation.”
Reviewer 2 Report
Dear authors,
the submitted manuscript provides interesting insight into the use of convenience cooking products and their association with selected aspects. Overall the paper is written at a good level, however, some minor issues must be addressed:
- to add short paragraph about consumer behaviour in the market with convenience cooking products in section with Introduction
- in methodology please indicate based on which you have designed selected questions (according to which previous research design?)
-please define precisely based on which demographic characteristics is your research sample representative
-please describe which method of sampling was applied in your research (convenience sampling, probability sampling etc.)
-please extend practical implications
-if possible add in the conclusion your future studies based on existing limitations
Author Response
the submitted manuscript provides interesting insight into the use of convenience cooking products and their association with selected aspects. Overall the paper is written at a good level, however, some minor issues must be addressed:
- to add short paragraph about consumer behaviour in the market with convenience cooking products in section with Introduction
- We have edited to include “Consumer behaviour models also show other attitudinal drivers of convenience food consumption such as guilt [15], moral motivations [16], “convenience orientation” towards time and energy savings [17, 18] and time and budget perceptions. Conversely, health consciousness and enjoyment reduce engagement with convenience foods [18]. Recent trends in demonstrate the normalisation of convenience foods and thus their importance in dietary patterns [15, 19]. While convenience and commercial require little or no preparation, reducing the time needed to prepare meals [6], they are typically lower in nutritional quality and are more energy dense [2, 20].”
- in methodology please indicate based on which you have designed selected questions (according to which previous research design?)
- The scales are adopted from previous research as referenced. The statements for agreement are based on standard likert scales as for agreement/disagreement and a reference has now been added.
-please define precisely based on which demographic characteristics is your research sample representative
- We have added explanation that representative refers to the Australian population as per census data and an appropriate reference.
-please describe which method of sampling was applied in your research (convenience sampling, probability sampling etc.)
- Sampling was conducted by a market research panel provider (Qualtrics) – using probability sampling – we have edited to specify.
-please extend practical implications
- We have added to the conclusion “Furthermore, this research has a practical utility in that these products have the potential to be recommended by dietitians and other health professionals for consumers to utilise their current cooking skills and confidences when cooking in the kitchen. The knowledge generated here may also inform the future design of these products to appeal to both consumer needs and nutritional needs.”
-if possible add in the conclusion your future studies based on existing limitations
- To our final call for more research we have added “including longitudinal research and intervention trials to demonstrate causation.”